# Castration-Resistant Prostate Cancer: From Uncovered Resistance Mechanisms to Current Treatments

**DOI:** 10.3390/cancers15205047

**Published:** 2023-10-19

**Authors:** Thi Khanh Le, Quang Hieu Duong, Virginie Baylot, Christelle Fargette, Michael Baboudjian, Laurence Colleaux, David Taïeb, Palma Rocchi

**Affiliations:** 1Centre de Recherche en Cancérologie de Marseille—CRCM, Inserm UMR1068, CNRS UMR7258, Aix-Marseille University U105, 13009 Marseille, France; khanh.le-thi@inserm.fr (T.K.L.); quang-hieu.duong@inserm.fr (Q.H.D.); virginie.baylot@inserm.fr (V.B.); michael.baboudjian@outlook.fr (M.B.); david.taieb@ap-hm.fr (D.T.); 2European Center for Research in Medical Imaging (CERIMED), Aix-Marseille University, 13005 Marseille, France; christelle.fargette@gmail.com; 3Vietnam Academy of Science and Technology (VAST), University of Science and Technology of Hanoi (USTH), Hanoi 10000, Vietnam; 4Department of Nuclear Medicine, La Timone University Hospital, Aix-Marseille University, 13005 Marseille, France; 5Department of Urology AP-HM, Aix-Marseille University, 13005 Marseille, France; 6Faculté de Médecine Timone, INSERM, MMG, U1251, Aix-Marseille University, 13385 Marseille, France; laurence.colleaux@inserm.fr

**Keywords:** androgen signaling, castration-resistant prostate cancer, androgen deprivation therapy (ADT), chemotherapy, immunotherapy, PARP inhibitors, radionuclide therapy

## Abstract

**Simple Summary:**

Castration-resistant prostate cancer (CRPC) remains a significant medical challenge, even with recent advancements in diagnosis and treatment. To improve patient outcomes, it is important to understand the underlying mechanisms of resistance to treatments and develop new therapeutic approaches. This review provides a brief summary of the current knowledge on the mechanisms that contribute to CRPC progression, including both androgen receptor (AR)-dependent and AR-independent pathways. It also discusses approved and currently investigated treatment options to treat patients with CRPC, such as novel chemotherapies, radiation therapy, immunotherapy, PARP inhibitors, and potential combined therapeutic strategies.

**Abstract:**

Prostate cancer (PC) is the second most common cancer in men worldwide. Despite recent advances in diagnosis and treatment, castration-resistant prostate cancer (CRPC) remains a significant medical challenge. Prostate cancer cells can develop mechanisms to resist androgen deprivation therapy, such as AR overexpression, AR mutations, alterations in AR coregulators, increased steroidogenic signaling pathways, outlaw pathways, and bypass pathways. Various treatment options for CRPC exist, including androgen deprivation therapy, chemotherapy, immunotherapy, localized or systemic therapeutic radiation, and PARP inhibitors. However, more research is needed to combat CRPC effectively. Further investigation into the underlying mechanisms of the disease and the development of new therapeutic strategies will be crucial in improving patient outcomes. The present work summarizes the current knowledge regarding the underlying mechanisms that promote CRPC, including both AR-dependent and independent pathways. Additionally, we provide an overview of the currently approved therapeutic options for CRPC, with special emphasis on chemotherapy, radiation therapy, immunotherapy, PARP inhibitors, and potential combination strategies.

## 1. Introduction

Prostate cancer (PC) is the second most prevalent cancer in men worldwide, representing 15% of all cancers diagnosed in males [1]. In the US, it is the most frequently diagnosed cancer and the second leading cause of cancer deaths in men. An estimated 288,300 new PC cases and 34,700 deaths from the disease are predicted to occur in the US in 2023 [2]. Patients who have advanced PC initially respond very well to androgen deprivation therapy. However, the treatment finally selects cancer cells that relapse into androgen deprivation, leading to the emergence of castration-resistant prostate cancer (CRPC). Patients with metastatic CRPC (mCRPC) have reduced life expectancy, with a median overall survival rate of below 2 years. The current definition CRPC, according to the European Association of Urology (EAU), is based on biochemical and/or clinical parameters demonstrating cancer progression in a castration environment (testosterone level < 50 ng/dL or 1.7 nmol/L) and standards proposed by Prostate Cancer Working Group 3 (PCWG3) [3] and/or the Response Evaluation Criteria in Solid Tumors (RECIST 1.1) [4]. Accordingly, biological progression refers to a condition where there are three consecutive increases in PSA (Prostate Specific Antigen), each at least one week apart, resulting in two increases of at least 50% over the lowest point, and a PSA level greater than 2 ng/mL. Radiographic progression is defined by the appearance of at least two new lesions on a bone scan or the progression of a measurable lesion according to the RECIST [5].

Over the past decade, there has been significant improvement in the treatment options for CRPC. Several drugs that increase overall patient survival, including androgen-receptor (AR) signaling inhibitors (such as abiraterone acetate, enzalutamide, apalutamide, and darolutamide), and radiopharmaceutical therapies (like radium-223 and 177Lu-PSMA-617) have been approved [6]. Advancements in precision medicine have led to the discovery of distinct subtypes of prostate cancer and the identification of genetic changes that predict the effectiveness of specific treatments [7,8]. In this review, we will provide an overview of the current understanding of the mechanisms that lead to CRPC, including both AR-dependent and -independent pathways. This review also summarizes the therapeutic options currently available for treating CRPC.

## 2. Mechanisms Underlying Castration-Resistant Prostate Cancer

There are currently many treatment options available for CRPC patients; however, the responses to these therapies are still limited. Knowledge of the underlying mechanisms associated with the CRPC phenotype can help to identify new promising therapeutics.

### 2.1. Androgen Receptor Overexpression

Androgen Receptor (AR) overexpression enables the survival and proliferation of tumor cells in limited-androgen conditions during androgen suppression treatment. Elevated AR expression at both the mRNA and protein levels has been frequently observed in CRPC. Amplification of the AR has been observed in 70% of cases of CRPC and is associated with significantly increased AR mRNA expression [9]. David A. et al. reported the existence of an enhancer that is amplified in 81% of castration-resistant metastatic patients. This enhancer has the potential to increase the expression of the *AR* gene independently of AR locus amplification in response to first-line androgen deprivation therapy (ADT) [9]. It should be noted that AR amplification is only seen in prostate tumors that have been exposed to androgen deprivation, indicating that AR amplification is a consequence of hormone therapy [10]. Additionally, epigenetic modifications, such as DNA methylation, histone acetylation, and miRNA modulation, could lead to AR overexpression in CRPC (Figure 1) [11].

### 2.2. AR Mutations

*AR* gene mutations have been found in around 10–20% of CRPC cases, while they are rarely observed in localized PC (0–4%) [12,13]. Most recurrent AR mutations are found in the ligand-binding domain (LBD) and/or cofactor binding regions [14]. These alterations could decrease the specificity of the AR for its main ligand, androgen; however, they allow AR specificity to extend to other hormones, such as progesterone and estrogen [12,15,16,17,18]. In addition, they could also disrupt the effectiveness of AR antagonists, causing them to act as AR agonists and resulting in resistance to treatments [19]. It has been reported that the H875Y mutation in CWR22 causes cells to exhibit altered ligand specificity. This mutant AR has shown transcriptional activity in response to testicular androgens, similarly to the wild-type AR. However, it differs from the wild type in its additional activation by adrenal androgens, dehydroepiandrosterone, and the antiandrogen hydroxyflutamide, as well as by estradiol and progesterone [20]. Along with AR coding mutations, several AR splicing mutations, leading to the abnormal expression of splice variants such as AR3 (also called AR-V7), AR4, and AR5, which lack the ligand-binding domain, are frequently observed in CRPC [21]. These variants encode a shorter receptor that lacks the ligand-binding domain and thereby constitutively activates AR pathways [22].

### 2.3. Alteration in AR CoRegulators

As a transcription factor (TF), the AR cooperates with a variety of regulatory proteins to form a productive transcription complex. These coregulators can either enhance (coactivators) or depress (cosuppressors) the transcriptional activity of the AR, thereby modulating the expression of androgen-regulated genes. As a consequence, alteration in the expression of AR coregulators can give a survival advantage to the cancer cells during castration therapy, promoting CRPC progression [23]. Several well-documented AR coactivators, including SRC1, SRC2, SRC3, ARA70, PIAS1, and Tip60, interact with the AR and enhance its transcriptional activity [24]. Levels of TIF2 and SRC1 expression have been observed to rise alongside the expression of the AR during the progression of prostate cancer cells following androgen deprivation. A previous study has also shown that these proteins are overexpressed in the majority of CRPC samples. The overexpression of TIF2 enhances the transcriptional activation of the AR in response to weaker steroids, such as adrenal androgens (DHEA and androstenedione), in two cell lines carrying AR mutations: LNCaP (T877A) and CWR22 (H874Y) [25]. The steroid receptor coactivator-3 (SRC3), also called amplified-in-breast cancer-1 (AIB1), has been identified as a potent coactivator of the hormone-activated AR [26] and has been demonstrated to promote CRPC progression [27]. Conversely, two well-known corepressors of the AR, NCoR (nuclear receptor corepressor) and SMRT (silencing mediator for retinoid and thyroid hormone receptors), have been shown to interact with the AR and decrease its transcriptional activity triggered by dihydrotestosterone [28,29,30].

### 2.4. Increased Steroidogenic Signaling Pathways

Accumulating evidence has reported that PC cells could survive after castration therapy by regulating intracrine androgen synthesis within the prostate. A typical example is the 5α-reductase enzyme, which catalyzes the conversion of testosterone (the most abundant free androgen) to the higher-affinity ligand 5α-dihydrotestosterone (DHT). Upon ADT, intraprostatic testosterone and DHT levels do not decline as markedly as do serum levels after ADT, buffering tumor cells from the loss of testicular androgen [31]. Increased 5α-reductase enzyme levels result from a polymorphism where there is a substitution of a valine with a leucine at the codon 89 [32]. Intraprostatic androgens can also be synthesized from cholesterol or other molecular precursors, such as DHEA (dehydroepiandrosterone). Indeed, DHEA could be converted into androstenedione, a substrate for conversion to testosterone. Another report has shown that CRPC samples have higher levels of expression of a variety of enzymes involved in de novo steroid synthesis, such as FASN, CYP17A1, CYP19A1, and UGT2B17, compared to primary PC [33].

### 2.5. Outlaw Pathways

In addition of being activated primarily by endogenous androgen ligands, the AR protein can also be stimulated through ligand-independent mechanisms, which are known as outlaw pathways. CRPC can thus be induced by the interactions between the cytosolic AR and many molecules such as different growth factors, cytokines, and kinases. Several growth factors, such as insulin-like growth factor-I (IGF-I), keratinocyte growth factor (KGF) and the epidermal growth factor (EGF), enable the stimulation of AR transcription activity at limited androgen levels or even in the absence of androgens [17]. Particularly, IGF1 has been reported to interact with the AR, facilitate AR translocation, and modulate the transcription of androgen-responsive genes. Moreover, it has been demonstrated to upregulate the expression of the AR coactivator TIF2 [34]. Along with growth factors, many cytokines, such as IL-6 and IL-8, have proven their ability to activate AR signaling in a ligand-independent manner [35]. Additionally, receptor tyrosine kinases (RTK) such as HER2/ERBB2 have been shown to be overexpressed in CRPC and restore AR signaling in low-androgen-concentration conditions [25]. 

### 2.6. Non-AR-Related Pathways: Bypass Pathways

While the mechanisms described above rely on AR transactivation to promote CRPC progression, alternative survival and growth pathways that are independent of AR activation can also drive androgen deprivation resistance. Of note, many outlaw pathways can also drive CRPC progression in an AR-signaling independent manner. High levels of serum insulin-like growth factor-I (IGF-I) and its receptor (IGF-1R) have been increasingly recognized to play a key role in prostate cancer progression. Binding of IGF-I with IGF-IR enables the stimulation of the transduction of a signal cascade, which, in turn, activates the expression of genes responsible for cellular growth and proliferation [10,23]. The upregulation of Akt pathway has been reported in a variety of human malignancies, including prostate cancer [36]. Indeed, AKT plays a central role in antiapoptotic pathways by phosphorylating and depressing Bcl2-associated agonist of cell death (BAD), a proapoptotic member of the BCL-2 protein family, and pro-caspase 9 [37]. AKT also promotes cellular proliferation by decreasing the expression of p27, a cell cycle inhibitor [38]. Several reports have shown that loss of PTEN, a tumor suppressor that inhibits PI3K/AKT signaling, frequently occurs in CRPC samples [39]. 

In addition, the heat-shock protein Hsp27 has previously been demonstrated to drive therapy resistance in PC, and its inhibitors, such as Hsp27 ASO (OGX-427) and small interference RNA (siRNA), have shown the ability to enhance chemotherapy. More recently, we have shown that Hsp27 plays a role in promoting the development of CRPC through its chaperone activity by selectively protecting some partner proteins involved in castration resistance, such as eIF4E, TCTP, Menin, and DDX5, from their ubiquitin–proteasome degradation [40,41,42,43]. CRPC has the potential to develop resistance to the antiandrogen enzalutamide by bypassing the AR blockade, which occurs due to increased activity of the glucocorticoid receptor (GR) [44]. Recent studies have indicated that heightened activity of the GR has been linked to the emergence of resistance to antiandrogen therapies in different clinical and preclinical models [44,45]. These findings could possibly be explained by the fact that GR expression is downregulated by AR signaling [46]. Additionally, the GR interacts with and mediates a subset of AR targets [44].

### 2.7. Evolution of t-NEPC from CRPC

During the initial diagnosis of PC, neuroendocrine prostate cancer (NEPC) is rare and represents only 0.5–2% of all PC cases. However, a larger proportion of treatment-induced neuroendocrine prostate cancer (t-NEPC) has been reported. Recent studies have indicated that approximately 17–30% of CRPC patients, following ADT and other treatments, may experience progression to t-NEPC [47,48]. The majority of evidence suggests that the origin of tNEPC is the transdifferentiation of adenocarcinoma cells into NEPC cells in response to different therapies, including ADT [47,48,49]. The transition to the NEPC phenotype following ADT treatments has been demonstrated in various preclinical models, including cell lines, genetically engineered mouse (GEM), and patient-derived xenografts [50,51,52]. In addition to ADT, other therapies, such as radiation and chemotherapy, have been reported to stimulate the NEPC phenotype [53,54,55]. Moreover, genetic and epigenetic alterations and tumor microenvironments (TMEs) containing a variety of cells and factors could support NEPC transition in CRPC patients [56,57,58,59]. Due to the absence of AR and PSA expression (AR-/PSA-), t-NEPC patients have shown limited responses to existing treatment options. To date, several therapeutics have been proposed for patients with t-NEPC, including cisplatin or carboplatin combined with etoposide or docetaxel.

## 3. Treatment Options for CRPC

Treatment options depend on several key factors, such as age, health condition, disease stage and grade, responses to previous therapies, and the potential advantages and side effects of the treatment. Besides ADT, other standard therapeutics for CRPC include chemotherapy, immunotherapy, radiation therapy, and PARP inhibitors (Figure 2).

### 3.1. Androgen Deprivation Therapy (ADT)

Androgens play a key role in regulating the proliferation and survival of prostatic cells. Androgen deprivation therapy aims to decrease the androgen levels produced by the testicular area, thereby preventing them from fueling prostate cancer cells, making the cancer cells shrink or grow slowly. ADT can be achieved either with surgery (pulpectomy) or pharmacologically.

For medical castration, a variety of compounds can be used, such as LHRH agonists and LHRH antagonists. Luteinizing hormone-releasing hormone (LHRH) agonists, which are also named LHRH analogs or gonadotropin-releasing hormone (GnRH) agonists, including leuprolide (Lupron^®^, Eligard^®^), goserelin (Zoladex^®^), triptorelin (Trelstar^®^), and histrelin (Vantas^®^). An approved LHRH antagonist for PC treatment is degarelix (Firmagon^®^) [60]. The luteinizing hormone (LH), which is made by the pituitary gland, transmits a key signal for testosterone production in the testes. Another oral ADT drug, relugolix (ORGOVYX™), has been approved by the FDA for patients with advanced PC [61]. A phase III clinical trial of relugolix (NCT03085095), in men with advanced PC with a 54% lower risk of serious adverse cardiovascular events, showed that it suppressed testosterone levels more quickly and effectively than leuprolide [62]. These drugs enable decreased LH production, thereby lowering the gonadal production of testosterone [63].

Currently, it is highly recommended to combine ADT with androgen synthesis inhibitors (e.g., abiraterone acetat) or with antiandrogen therapies (e.g., enzalutamide, apalutamide, darolutamide) as well as in combination with chemotherapies (docetaxel) [64]. The guidance from the European Association of Urology (EAU), the American Urological Association (AUA), and the National Comprehensive Cancer Network (NCCN) indicates that ADT should be maintained for CRPC patients for several reasons [65,66,67]. In fact, even serum testosterone levels are suppressed significantly by ADTs; this hormone still remains at low levels in the serum, thereby maintaining androgen functions in the prostate and in the prostate tumor microenvironment [68]. Moreover, the maintenance of intratumor androgens and of androgen-responsive genes including the AR and PSA has been previously reported [33,69]. Previous sections mentioned adaptive mechanisms of androgen signaling that contribute to CRPC, including overexpression, mutation, and splice variations of the AR, as well as changes in androgen synthesis pathways. The presence of androgens or AR ligands enables these mechanisms to persist or even intensify, which hinders the effectiveness of other treatments [70].

### 3.2. Chemotherapy

Different chemotherapeutic compounds have been approved for PC treatment, including taxane (docetaxel (Taxotere^®^) and cabazitaxel (Jevtana^®^), mitoxantrone (Novantrone^®^), and estramustine (Emcyt^®^). In addition, several other chemotherapy drugs, such as cisplatin, oxaliplatin, and carboplatin, are currently in clinical evaluation for their utilization in PC [71]. Both docetaxel and cabazitaxel are drugs that stabilize microtubules, leading to mitotic arrest and apoptotic cell death. In addition, these drugs act on AR nuclear translocation that depends on intact and dynamic microtubules [72]. Docetaxel and cabazitaxel are delivered intravenously once every three weeks for around 10 cycles [73]. In most cases, docetaxel is used as the first-line chemo treatment and cabazitaxel is considered as a second-line treatment to overcome docetaxel resistance [74].

Another chemotherapy drug, mitoxantrone, was approved in 1996 by the FDA to treat CRPC patients. In a phase III trial, 161 patients with symptomatic mCRPC were randomly assigned to receive mitoxantrone plus prednisone versus prednisone alone. The primary study outcome, pain relief response, was achieved in 29% of the patients treated with mitoxantrone versus 12% of the patients treated with prednisone, with no survival benefit observed [75]. Mitoxantrone could cause DNA damage by generating DNA breaks through its binding-to-DNA sequence. It could also act as a topoisomerase II inhibitor, consequently disrupting the DNA replication process [76].

Chemotherapies target rapidly dividing cells to fight cancer, but other constantly fast-dividing cells, like those in the bone marrow, mouth, and intestines, as well as hair follicles, can also be affected, causing important side effects. The type, dose, and duration of chemotherapy determine the undesirable effects. Common ones include hair loss, mouth sores, loss of appetite, nausea, vomiting, diarrhea, increased infection risk, easy bruising or bleeding, and fatigue. These side effects typically disappear after treatment, and some drugs can help to manage them [77].

### 3.3. Immunotherapy

The purpose of immunotherapies is to boost patient’s own immune system to combat cancer cells. To date, three immunotherapies have been approved by the FDA for the treatment of patients with CRPC, including an immune-cell-based vaccine and two immune checkpoints inhibitors (ICI) [78]. Prostate cancer cells express several proteins, including PSA and prostatic acid phosphatase (PAP), that have been identified as potential targets for antigen-based vaccines for advanced prostate cancer [79]. Sipuleucel-T (first approved 29 April 2010), the first FDA-approved cellular immunotherapy, is produced by collecting dendritic cells from patients. These cells are then activated by incubating them ex vivo with PA2024, a recombinant PAP target fused with the granulocyte–macrophage colony-stimulating factor (GM-CSF). Finally, the activated dendritic cells are reinjected into patients, serving as an autologous cellular vaccine that elicits T-cell immune responses targeting the PAP antigen [80] (Figure 2B). Sipuleucel-T has been tested in several phase III clinical trials (NCT00065442, NCT00005947, and NCT01133704) and received US FDA approval (Provenge^®^; Dendreon, Inc.; Seattle, WA, USA) [81,82]. In the IMPACT trial (NCT00065442), the median survival of patients treated with Sipuleucel-T was 25.8 months, compared with 21.7 months for patients treated with the placebo, and no difference in the time to objective disease progression was observed [83].

The two other FDA-approved immunotherapies for advanced solid tumors including CRPC are pembrolizumab (first approved 4 September 2014) and dostarlimab (first approved 22 April 2021). They are both ICIs in the form of monoclonal antibodies (mAbs). These drugs inhibit the programmed death 1 pathway (PD-1) by blocking the interaction between the PD-1 receptor and its ligands (PD-L1 and PD-L2), which in turn activates the antitumor immune response [84]. Pembrolizumab and dostarlimab have been approved for patients with unresectable or metastatic microsatellite instability-high or mismatch repair-deficient (dMMR) tumors [78,85]. A CRPC patient who was resistant to abiraterone, docetaxel, and cabazitaxel reportedly experienced a significant decrease in PSA levels, as well as reductions in tumor size and the metastatic lymph nodes, after undergoing pembrolizumab therapy [86]. Furthermore, a phase II single-arm trial (NCT02312557) found that pembrolizumab in combination with enzalutamide provides deep and durable responses for mCRPC patients, regardless of tumor PD-L1 expression or DNA repair defects [87].

Immunotherapies like sipuleucel-T and ICIs offer an appealing alternative to CRPC standard treatments such as chemo- and hormone therapy. However, the benefit of immunotherapies in patients with CRPC remains modest. These observations may be the result of the complexity and heterogeneity of the PC microenvironment. Clearly, a better understanding of the mechanisms underlying immune escape in PC is needed to find which immunotherapeutic targets could be beneficial for the patients and identify the optimal timing to introduce these compounds [88]. This includes identifying and targeting specific antigens on prostate cancer cells, as well as finding ways to overcome the immunosuppressive tumor microenvironment. Additionally, optimal combinations between immunotherapies and standard anti-cancer drugs to treat CRPC are still under investigation.

### 3.4. Radiation Therapy

#### 3.4.1. PSMA-Targeting Radioligand Therapy

The most efficient and recent strategy for CRPC treatment relies on the use of a prostate-specific membrane antigen (PSMA)-targeting radiolabeled molecule. The PSMA is moderately expressed in several normal tissues. However, its expression is considerably elevated in prostate cancer, regardless of the stage of the disease [89]. As 90% of CRPC cases overexpress the PSMA, they can be treated with PSMA radioligand therapy (RLT). In addition, PSMA also has pro-proliferative functions as the cleavage of glutamate by the PSMA activates the PI3K/AKT/mTOR pathways. High expressions of PSMA are associated with poor clinical outcomes and may be involved in the mechanism by which PC tumors acquire resistance [90,91].

The principle of PSMA RLT is to selectively deliver radiation to cancer cells through the systemic administration of PSMA radioligands such as PSMA-617 [92]. The PSMA RLT scheme relies on regular cycles (every 6 weeks) of fixed activity (7.4 GBq/cycle), similarly to chemotherapy. Dosimetry is not often needed, but it can be mandatory for high-dose therapies. The first step is to select patients that are likely to benefit from this treatment. Patients should be positive for ^68^Ga- or ^18^F-PMSA PET (positron emission tomography) scans, both of which act as companion diagnostic tools. In the TheraP trials (NCT03392428), approximately 28% of patients were excluded due to various criteria. These criteria included imaging discrepancies between the results of the ^18^F-FDG PET and the PSMA PET, as well as low/absent tumor uptake [93]. Lutetium-177 PSMA-617 has been approved by the FDA and EMA for CRPC treatment in patients that have progressed under ASI and one line of taxane-based chemotherapy, based on the excellent results of the VISION trial (NCT03511664). It is expected that promising results before chemotherapy will promote this therapy even earlier, taking into account that such patients will benefit from an earlier triplet that includes chemotherapy [94,95,96,97]. There are many ongoing studies using PSMA RLT, such as PSMAfore (NCT04689828); SPLASH (NCT04647526); and ECLIPSE (NCT05204927). Several mechanisms of resistance to PSMA RLT have been described, such as heterogeneity of dose distribution not covered by the cross-fire effect of ^177^Lu, visceral organ involvement that is less controlled by radiation, an insufficient cytotoxic effect of ^177^Lu, and a tumor mutational burden that confers a resistance to radiation damages [98].

Various areas of improvement for ^177^Lu-PSMA RLT are currently being evaluated: (1) pharmacological manipulation of targets (also called phenotype adjustment) with AR signaling inhibitors (e.g., enzalutamide) [99], (2) validation of image-derived biomarkers for the prediction of responses and prognosis [100,101,102], and (3) enhancement of radiation-induced cell death by combination strategies with immune checkpoint inhibitors or PARP inhibitors [103]. To achieve the intended goal, radiation-induced mutations should involve coding DNA regions in order to induce protein misfolding and the subsequent immunogenic response, (4) replacing ^177^Lu with alpha emitters such as actinium-225 (NCT04597411) [104,104]. ^225^Ac-PSMA-617 has been shown to be a viable treatment option with PSA responses in the majority of cases, a finding associated with increased overall survival (OS). It efficacy is, however, decreased in heavily pretreated patients. Substantial bone marrow (one-third) and salivary gland (almost constant) toxicity profiles have been reported but remain tolerable. We urgently need prospective comparative data on actinium–PSMA activity and safety. It is nowadays impossible to establish Ac-225-PSMA as a preferred salvage treatment option for disease progression following ^177^Lu-PSMA [105]. A significant limitation of PSMA-targeted therapy is the absence of consistent PSMA expression across different metastatic tissues in a patient, especially in the liver, sometimes related to an NE (neuroendocrine) differentiation [106,107,108]. Furthermore, during prostate cancer progression, many patients undergo declines in PSMA expression, making them ineligible for the initiation or continuation of PSMA-directed therapies [97].

#### 3.4.2. Radium-223 Dichloride

Radium-223 dichloride (223Ra; Xofigo^®^) received the FDA approval for the treatment of bone pain in patients with mCRPC based on the 2013 ALSYMPCA study. When radium-223 atoms decay, they emit four alpha particles that are high-energy and capable of damaging DNA strands, thereby killing cancer cells (Figure 2C). Radium-223 mimics calcium and forms complexes with hydroxyapatite, a mineral found in areas of high bone turnover. This allows it to specifically target bone metastasis [109,110].

The alpha particles emitted by Radium-223 have a short range and a high linear energy transfer. This allows them to kill a small area of tissue intensely (tumors) while sparing most of the normal bone tissue. This is a significant advantage over other radiation therapy options, as it represents a targeted therapeutic approach that minimizes damages in healthy cells. Radium-223 has a dual effect of reducing pathological bone turnover and irradiating tumors. The reduction in pathological bone turnover is especially important for patients with mCRPC, as this condition often leads to the development of bone metastasis, which can cause significant pain and other complications. Radium-223 has been demonstrated to prolong OS in symptomatic patients who have suffered from multiple-bone-metastatic CRPC, without visceral or nodal involvement [111]. CRPC patients with bone metastasis and treated with 223Ra demonstrated significant improved OS in phase III clinical trials [112].

### 3.5. PARP Inhibitors

The reason for using PARP inhibitors (PARPis) is primarily the high occurrence of genetic mutations in prostate cancer and the concept of synthetic lethality, where the combination of two deficiencies leads to cell death [113]. PARP enzymes participate to DNA repair pathways, specifically in the repair of single-strand breaks via base excision repair, in both normal and cancer cells. The inhibition of PARP enzymes stimulates the conversion of single-strand DNA breaks (SSBs) into double-strand DNA breaks (DSBs) during DNA replication. In normal cells, various DNA damage response (DDR) pathways are activated to repair DSBs, including those of non-homologous end joining (NHEJ), microhomology-mediated end joining (MMEJ), and, most commonly, homologous recombination repair (HRR) [114]. However, in cancer cells with HRR deficiency (HRD), treatment using a PARP inhibitor will ultimately cause buildup of DSBs and result in cell death (Figure 2). Additionally, a PARPi could exert its anticancer activity through its ability to trap PARP1 and PARP2 on damaged DNA sites, which would eventually hinder DNA repair, replication, and transcription [6]. Furthermore, a recent study analyzed the genomic landscape of mCRPC in 197 patients using whole-genome sequencing (WGS). This analysis identified eight genomic subgroups in mCRPC, with 11.2% (22/197) of patients showing significant features of homologous recombination deficiency (HRD), mainly due to biallelic BRCA2 inactivation. Within this subgroup, seven out of twenty-two did not have biallelic BRCA2 inactivation, but four of them had at least one (deleterious) aberration in other BRCAness-related genes [8].

Four PARPis (olaparib, rucaparib, niraparib, and talazoparib) have shown significant antitumor activity against mCRPC. Currently, the FDA has approved two PARPis (olaparib and rucaparib) as monotherapies for metastatic castration-resistant prostate cancer patients [115]. Based on the PROfound study findings, olaparib was approved in 2020 for treatment of mCRPC patients who carry alterations in the genes involved in homologous recombination repair and have experienced progression after androgen receptor signaling inhibitor (ARSI) therapy [116]. Recently, a clinical trial concluded that mCRPC patients with BRCA mutations and who were treated with rucaparib could significantly prolong their progression-free survival time compared to the patients who received the control medication [117]. Furthermore, in May 2023, the FDA approved the first PARPi-based combination for mCRPC patients based on the results of the PROpel trial (NCT03732820). This treatment involves using olaparib, abiraterone, and prednisone as the initial treatment for BRCA-mutated mCRPC patients [118]. Niraparib and talazoparib have only been evaluated as monotherapies in mCRPC patients with DNA repair alterations in phase II trials [119,120]. However, combinations based on these drugs have been authorized for mCRPC patients. The TALAPRO-2 study and the MAGNITUDE trial showed positive results for the combination therapies of talazoparib with enzalutamide and niraparib with abiraterone plus prednisone, respectively, leading to their approval [121,122]. There are currently a number of clinical trials that currently examine the effectiveness of novel PARP inhibitors either as monotherapies or in combination with other agents. These trials, in addition to the PARP inhibitors that have already been approved, offer encouraging prospects for new treatment options for patients with metastatic prostate cancer [123].

## 4. Conclusions

Although extensive research has been conducted on the mechanisms driving therapy resistance in CRPC, treating this disease remains a significant medical challenge. Therefore, it is crucial to continue studying this theme to identify more effective treatments. Currently, various strategies are used to treat CRPC patients, including maintaining ADT, taxane-based chemotherapy, therapeutic nuclear medicine, the selective use of external beam radiotherapy, PARP inhibitors, and immunotherapy. During treatment, preserving patients’ quality of life (QoL) and bone marrow reserves is of utmost importance. Combining therapies could open up new perspectives.

## Figures and Tables

**Figure 1 cancers-15-05047-f001:**
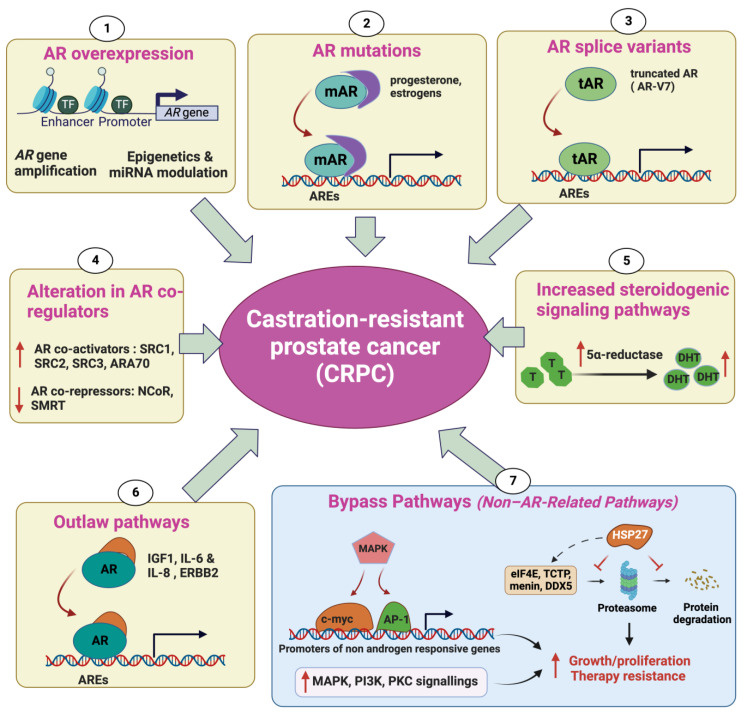
Mechanisms of castration-resistant prostate cancer (CRPC). AR overexpression enables the survival and proliferation of tumor cells in limited-androgen conditions during androgen suppression treatment. A high AR expression level can be due to *AR* gene amplification, epigenetics, and miRNA modulation (1). Point mutations in the ligand-binding domain of the *AR* gene lead to the increased affinity of the mutated AR (mAR) to other hormones, such as progesterone and estrogen, thereby modulating androgen-responsive gene transcription independently from androgen (2). The emergence of AR-splicing isoforms, such as AR3 (also called AR-V7), AR4, and AR5, encoding the truncated AR protein (tAR), which lacks the ligand-binding domain, results in constitutive activation of the AR, thereby promoting variant-carrying tumor cells to ignore the need for androgen (3). Overexpression of AR coactivators and the decreased expression of AR corepressors will result in an increase in AR-regulated transcription (4). Increased production of 5α-reductase can provide sufficient androgens for AR activation in cancer cells (5). CRPC can be induced by outlaw pathways in which AR signaling can be activated in a ligand-independent manner by other molecules than androgens, such as growth factors, cytokines, and kinases (6). The bypass pathways, involved in CRPC progression, increase the activity of MAPK, PI3K, and PCK cascades, leading to either the stimulation of alternative growth pathways or the enhancement of survival signaling independently from AR signaling. Alternative pathways also involve the overexpression of heat-shock protein 27 (HSP27), which mediates its cytoprotective function by protecting its interacting proteins (eIF4E, TCTP, Menin, DDX5) from their degradation by the proteasome (7) (↑ in red: increase, ↓in red: decrease).

**Figure 2 cancers-15-05047-f002:**
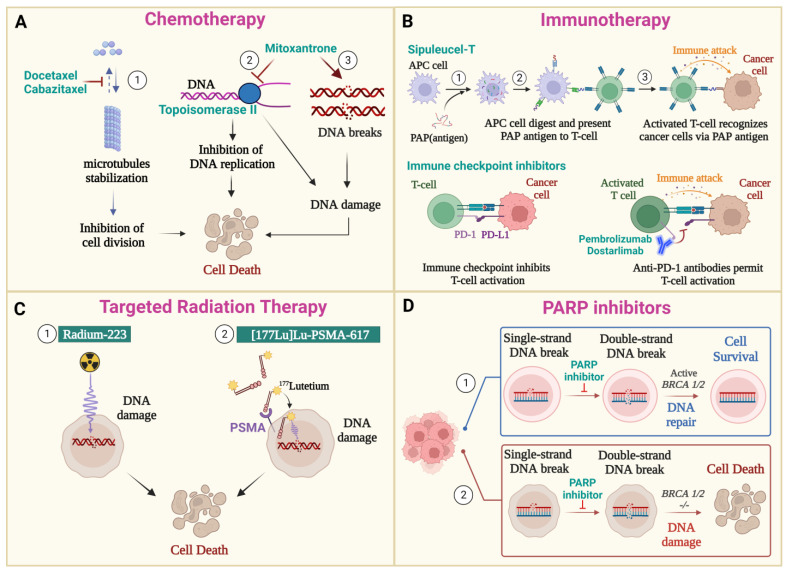
Current therapies for castration-resistant prostate cancer (CRPC) treatment. Along with androgen deprivation therapy (ADT), the approved therapies by the FDA and EMA to treat CRPC include chemotherapy (A), immunotherapy (B), radiotherapy (C), and PARP inhibitors (D). (**A**) **Chemotherapy:** (1) Docetaxel and cabazitaxel prevent microtube depolymerization, therefore inhibiting cell division and causing cell death. Mitoxantrone could induce cell death via the inhibition of topoisomerase II (2) and DNA damage induction (3). (**B**) **Immunotherapy:** Sipuleucel-T stimulates the T-cell anti-tumor activity by targeting the prostatic acid phosphatase (PAP) protein—an overexpressed protein in prostate cancer cells. (1) Antigen-presenting cells (APCs) isolated from patients will digest PAP proteins into small peptides and display these PAP peptides on their surfaces. (2) APCs present PAP peptides to T-cells, which can then recognize cancer cells that express PAP on their surfaces and activate immune cytotoxic effects to kill CRPC cells (3). Immune checkpoint inhibitors, including pembrolizumab and dostarlimab, are utilized to combat cancer cells by blocking immune checkpoint pathways, which reactivates cytotoxic lymphocytes antitumor responses. These medications work by preventing interaction between the PD-1 receptor and its ligands (PD-L1 and PD-L2), consequently inhibiting the programmed death 1 pathway and triggering the immune response against cancer cells. (**C**) **Targeted Radiation Therapy:** (1) Radium-223 induces cell death by generating DNA damage via emitting high-energy α-particles; (2) [^177^Lutetium]-PSMA-617 contains the ligand of the prostate-specific membrane antigen of prostate cancer cells (PSMA-617) conjugated with radiolabeled ^177^Lutetium. This drug causes DNA damage via the release of Ɓ and γ particles, thus leading to cell death. (**D**) **PARP Inhibitors:** The inhibition of PARP enzymes leads to double-strand DNA breaks from single-strand DNA breaks (SSBs). (1) The *BRCA1/2* gene will be activated to repair DSBs and promote cell survival. (2) Cells with *BRCA1/2* alterations cannot repair DSBs, leading to DNA damage and inducing cell death.

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
