# Peer review of "Castration-Resistant Prostate Cancer: From Uncovered Resistance Mechanisms to Current Treatments"

_cancers, 2023, doi:10.3390/cancers15205047_

Round 1
Reviewer 1 Report
This is a very comprehensive review of the current status of prostate cancer treatment. The figure is well described and will benefit the reader of cancers.
I have only two minor comments.
Minor
2.5. PARP inhibitors.
Please include on going study of NCT04821622 for the Tarazoparib study in mHSPC. This could be the new direction.
2.4.1. PSMA- targeted radioligand therapy
Please include the data that 28% of patients were excluded from the study due to the discordance between Ga PSMA and FDG PET on TheraP Study NCT03392428.
The heterogeneity of PSMA expression is the major limitation of PSMA PET, as described https://pubmed.ncbi.nlm.nih.gov/37038004/, please discuss the heterogeneity of the PSMA expression, like liver mets and NEPC.
Author Response
"Please see the attachment."

Reviewer 2 Report
It is a nice review that clearly summarizes the canonical mechanisms of CRPC development and ongoing therapies. Recent studies in this field focus on the evolution of CRPC to NEPC. Generally, the development to NEPC is a major challenge in current CRPC treatment. It would be better to include the mechanisms by which CRPC to NEPC transition. Androgen–glucocorticoid interaction is another mechanism underlying CRPC that is missing in the current manuscript.
Other minor comments:
- Line 5, Line 281 typo or format mistakes
- Line 321, it is confusing why the gene alternations of PTEN were listed here, which is not directly associated with DNA damage repair pathways.
Author Response
"Please see the attachment."

Reviewer 3 Report
The manuscript by Le at al., clearly depicts the current knowledge on the resistance mechanisms and the current treatment options. However at present the review need to rechecked for the fact. In several occasions the description not fully captures the real fact. Some are given below and many are not. The authors are advised to go through the manuscript in full and verify the fact form original publications and not form back reference and abstract.
1. Update the CRPC definition as per the PCWG3 consensus as PCWG3 includes the PCWG2 and later knowledge.
2. In patients with CRPC, treatment options remains limited. Knowledge of the underlying mechanisms associated with CRPC phenotype can help to identify new treatment options. – please change the sentence to capture current scenario – Treatment options are not limited – however the limitation is due to the treatment response.
3. Abstract should be rewritten by capturing the overall theme and summary of the review
4. A short introduction will help the readers
5. Information corresponding to Ref 4 need to verified for the fact. The fact is that AR amplification is about 65-70 in CRPC.
6. Fact check – “These alterations decrease specificity of AR for its main ligand androgen; however, such mutations allow to extent AR specificity to other hormones such as progesterone, oestrogens [7,9–12]” – check it again AR mutation also results in even minimal androgens is enough to drive AR signaling and mutation also leads cause AR antagonists to turn into agonists (eg., nilutamide).
7. SRC-1-3 are the major regulators and known to activate AR signaling. There is no mention of it in section 1.3.
8. Figure 2 did not capture PDL1 targets.
Not applciatble
Author Response
"Please see the attachment."

Round 2
Reviewer 3 Report
The revised version substantially addressed the previous concerns